# Sleep and Microbiome in Psychiatric Diseases

**DOI:** 10.3390/nu12082198

**Published:** 2020-07-23

**Authors:** Jolana Wagner-Skacel, Nina Dalkner, Sabrina Moerkl, Kathrin Kreuzer, Aitak Farzi, Sonja Lackner, Annamaria Painold, Eva Z. Reininghaus, Mary I. Butler, Susanne Bengesser

**Affiliations:** 1Department of Medical Psychology, Medical University of Graz (MUG), 8036 Graz, Austria; jolana.wagner-skacel@medunigraz.at; 2Department of Psychiatry and Psychotherapeutic Medicine, Medical University of Graz (MUG), 8036 Graz, Austria; nina.dalkner@medunigraz.at (N.D.); sabrina.moerkl@medunigraz.at (S.M.); kathrin.kreuzer@medunigraz.at (K.K.); Annamaria.painold@klinikum-graz.at (A.P.); eva.reininghaus@medunigraz.at (E.Z.R.); 3Otto Loewi Research Center (for Vascular Biology, Immunology and Inflammation), Division of Pharmacology, Medical University of Graz (MUG), 8036 Graz, Austria; aitak.farzi@medunigraz.at; 4Otto Loewi Research Center (for Vascular Biology, Immunology andI), Division of Immunology and Pathophysiology, Medical University of Graz (MUG), 8036 Graz, Austria; sonja.lackner@medunigraz.at; 5Department of Psychiatry, University College Cork, T12 YN60 Cork, Ireland; mary.butler@ucc.ie

**Keywords:** sleep, gut microbiome, gut microbiota circadian rhythms, nutrition, psychiatric diseases

## Abstract

Objectives: Disturbances in the gut–brain barrier play an essential role in the development of mental disorders. There is considerable evidence showing that the gut microbiome not only affects digestive, metabolic and immune functions of the host but also regulates host sleep and mental states through the microbiota–gut–brain axis. The present review summarizes the role of the gut microbiome in the context of circadian rhythms, nutrition and sleep in psychiatric disorders. Methods: A PubMed search (studies published between April 2015–April 2020) was conducted with the keywords: “sleep, microbiome and psychiatry”; “sleep, microbiome and depression”; “sleep, microbiome and bipolar disorder”, “sleep, microbiome and schizophrenia”, “sleep, microbiome and anorexia nervosa”, “sleep, microbiome and substance use disorder”, “sleep, microbiome and anxiety”; “clock gene expression and microbiome”, “clock gene expression and nutrition”. Only studies investigating the relationship between sleep and microbiome in psychiatric patients were included in the review. Results: Search results yielded two cross-sectional studies analyzing sleep and gut microbiome in 154 individuals with bipolar disorder and one interventional study analyzing the effect of fecal microbiota transplantation in 17 individuals with irritable bowel syndrome on sleep. In patients with bipolar disorder, *Faecalibacterium* was significantly associated with improved sleep quality scores and a significant correlation between *Lactobacillus* counts and sleep. Conclusion: Translational research on this important field is limited and further investigation of the bidirectional pathways on sleep and the gut microbiome in mood disorders is warranted.

## 1. Introduction

### 1.1. Background

Sleep is not only necessary for recovery from physical and mental illness, but is also essential for a range of brain functions such as neural cell growth, synaptogenesis and memory function [1]. In a recent meta-analysis, insomnia was identified as an important predictor for the onset of depression, anxiety, alcohol abuse and psychosis representing the vital importance of sleep for mental wellbeing [2]. Nevertheless, the molecular underpinning of sleep is still not completely revealed and little is known about the bidirectional relationship between the microbiota–gut–brain axis (MGBA) and sleep in psychiatric disorders. The current review will give an overview on the neurobiological background of the circadian system and the MGBA in the context of sleep and psychiatric disorders with the focus on chronobiology and biochemistry.

### 1.2. The Molecular 24-h Clock

All organisms on Earth have developed an inner circadian (24-h) clock, which regulates important body functions such as heartbeat, blood pressure, hormone secretion (e.g., cortisol, melatonin) and in particular, mood states, via transcription of a network of hundreds of clock gene controlled genes (CCGs). Disruption of sleep and circadian rhythmicity is a core feature of mood disorders, which was demonstrated in a UK study with 91,105 study participants published in LANCET Psychiatry [3]. Multiple clock gene variants predispose to psychiatric disorders such as major depressive disorder (MDD), bipolar disorder (BD), attention deficit hyperactivity disorder (ADHD), schizophrenia and delirium. The nominal association between clock gene variants and affective disorders (MDD and BD) have been identified already by early gene-association and gene expression studies [4,5,6,7,8,9,10,11,12,13,14,15]. Also ADHD was associated with gene variants (e.g., rs1801260) in gene-association studies [16,17,18]. A post mortem gene expression study of the human dorsolateral prefrontal cortex by Seney et al. also showed that the diurnal rhythm in gene expression is different in subjects suffering from schizophrenia compared to the healthy control group [19]. Aside from the role of the circadian clock in psychiatric disorders, many cardiovascular functions such as blood pressure, heart rate, endothelial function and thrombus formation are influenced by a circadian rhythm. It is noteworthy that the onset of acute myocardial infarction, stroke and arrhythmias is also linked to the circadian clock [20] and various single nucleotide polymorphisms (SNPs) of the clock genes *ARNTL (Aryl Hydrocarbon Receptor Nuclear Translocator Like. Syn.BMAL1)* and *CLOCK* (circadian locomotor output cycles kaput) are associated with type 2 diabetes mellitus [21].

It is unsurprising that the molecular clock plays a crucial role in psychiatric disorders, given that mood and circadian rhythms are closely linked. Clinical experience shows that disturbed circadian rhythms provoked by jetlag or reduced sleep can trigger mood swings and sleep disorders [22,23,24]. The clock gene network, which is present in every living organism, operates over a period of 24 h in human beings. The 24-h clock is reset and synchronized every day by light, which is received by the retina. The light impulse is transmitted to the suprachiasmatic nucleus (SCN) in the hypothalamus by the retinohypothalamic tract [25]. Each morning the transcription of cryptochrome (*CRY1-2*) and period (*PER1-3*) genes is activated by activating gene products encoded by the clock genes *ARNTL* (*BMAL1*) and *CLOCK*. Cryptochrome and period genes become phosphorylated in the cytoplasm, where they accumulate over approximately 12 h. Then the phosphorylated, cycle length determining *CRY* and *PER* proteins in the cytoplasm enter the nucleus, where they suppress the activating function of *ARNTL* and *CLOCK*, which stops the transcription of further *CRY* and *PER* genes (negative feedback loop) [22]. Therefore, core clock genes manage the molecular 24-h clock by an interlocked transcriptional–translational feedback loop, which generates the circadian regulation of important body functions such as hormone release (melatonin, cortisol etc.), body temperature, heartbeat and sleep. Melatonin production is intrinsically linked to circadian rhythms, with levels gradually increasing in the two hours before natural sleep onset and peaking several hours later [26]. Melatonin regulates various physiological and neuroendocrine functions through activation of MT_1_, coupled G_i/o_-type proteins, and MT_2_ receptors coupled to G_q_-type proteins. However, the underlying regulatory mechanism of melatonin release in the SCN remains elusive [27]. Aside from sleep regulation itself, the clock gene circle can directly affect mood by impacting neurotransmitter breakdown. There is a strong positive correlation between the clock gene *ARNTL* and the gene *MAOA* (monoamine oxidase A). The clock gene products of *ARNTL* and *NPAS2* activate, as heterodimers, the transcription of the monoamine oxidase A (*MAOA*) gene. This has important consequences for the breakdown of neurotransmitters, which are metabolized by the enzyme *MAOA* and thus, for mood regulation [22,23,24]. Methylation of cytosines in regulatory regions of the promoter of *ARNTL* can affect *ARNTL’s* gene expression by silencing of the gene, which can hypothetically alter neurotransmitter breakdown. Previous results from Bengesser et al. showed that methylation of *ARNTL* differed significantly between bipolar disorder and controls [28]. Gene–environment interactions can affect gene expression regulation in the human body. As mentioned, hypermethylation of regulatory CpG islands around the promoter are associated with silencing of a gene. Methylation of cytosines in regulatory targets of DNA can be affected by diverse gene–environment-interactions such as chronic stress, aging, obesity, smoking, mood stabilizer intake, early life stress or trauma and nutrition [29,30]. Methylation of CpG islands (CG rich elements) can be favored by processed nutrients (e.g., polyphenols from green tea, cocoa or coffee) or by metabolites of gut-bacteria [31,32]. Thus, it is not surprising that there was a negative correlation between the bacterial diversity of the gut-microbiome and the methylation of *ARNTL* in fasting blood DNA of study participants with bipolar affective disorders [33]. Therefore, the gut-microbiome, which refers to the collection of genetic material of the gut microorganisms, affects the whole human organism including brain-health and sleep.

### 1.3. Gut Microbiome and Sleep

There is considerable evidence showing that the gut microbiome not only affects digestive, metabolic and immune function but also regulates sleep and mental states [34]. This is accomplished through the MGBA, in close interaction with emotions, physiological stress and circadian rhythms. Sleep patterns are affected by changes in intestinal permeability, immune system activation, inflammation, energy harvest and bacterial diversity [35,36]. The number of gut microbiota as well as the abundance of special species as *Bacteroidetes* and *Clostridia* oscillate during the light-dark cycle in mice [37]. Bacterial load is highest during their active phase with high abundance of *Bacteroidetes*, while the lowest bacterial load is observed during their rest phase with high abundance of the phylum Firmicutes [38,39]. Knockout of clock genes, including *ARNTL* as well as *PER1/2* attenuate these oscillations [37,40]. Voigt et al. found that core clock gene mutations caused gut microbiome dysbiosis and that this was exacerbated by dietary intestinal stimuli in mice [41]. The potential impact of dietary composition and nutrient intake on gut microbiota composition and mental health has already been reviewed comprehensively by others [42,43]. Briefly, the gut microbiota composition is, besides other environmental and individual determinants, strongly modulated by dietary composition and nutrient intake. In particular, non-digestible carbohydrates such as fermentable dietary fibers, resistant starch and polyphenols, have been shown to act prebiotically since they are utilized by the resident microorganisms of the large intestine. These dietary components are likely to enhance alpha diversity and influence gut microbiota composition beneficially by selectively stimulating the growth of certain microorganisms [42]. A probiotic is a living microorganism, which when present in adequate amounts confers a health benefit to the host, such as fermented food like yoghurt, kefir, tempeh, kimchi, kombucha and others Probiotic bacteria can be consumed in the form of commercially-produced dietary supplements such as capsules, tablets or fortified dietary products. Additionally, dietary supplements containing certain strains of bacteria belong to probiotics. Since beneficial effects of probiotics on mental health are likely, they are also referred to as “psychobiotics” [44]. As mentioned before, probiotics seem to affect epigenetic mechanisms of gene expression regulation by influencing the methylation of CG sites in DNA. Methylation of cytosines in regulatory sequences near the promoter, the starting point of transcription, leads to silencing of a gene. Silencing of a gene consequently leads to gene expression and protein translation changes, which affects networks in the human body. It has been found that probiotics affect pathways that are necessary to create and transfer the methyl-group, e.g., the one-carbon (C1) metabolism. Certain probiotic bacteria (*Lactobacillus helveticus R0052* and *Bifidobacterium longum R0175*) increased the donor for methyl-groups, S-adenosylmethionin (SAM), in an animal model of depression [45]. Methylation of DNA can also be affected by short chain fatty acids (SCFAs) such as butyrate, which are produced by gut-bacteria and can cross the blood–brain barrier. Also, nutrients such as folate and polyphenols from cocoa, green tea or certain fruit juices can affect epigenetic changes of DNA and consequently, gene expression and protein biosynthesis [31,32]. Thus, probiotics that alter microbiome diversity and, therefore, the gut-microbiome metabolites, have in theory the potential to positively influence gene expression regulation and the MGBA [31]. Nevertheless, to date transcriptomics- and metabolomics analyses before and after randomized, placebo-controlled, probiotics trials are to our best knowledge completely lacking in MDD, BD and psychotic disorders. There have only been a handful of studies carried out on healthy participants and patients with other medical disorders (e.g., autism, mastitis, inflammatory bowel diseases), which analyzed transcriptomics and metabolomics after probiotics trials [46,47,48,49,50,51].

Circadian clock misalignment, sleep deprivation and shift-work experience change circadian clock gene expression and microbial community structure [52]. Another important novel gut–brain axis mechanism is the modulation of gene expression in the human organism by interfering with epigenetic modulation of genes (i.e., through methylation) [53]. While the gut microbiota are affected by circadian signals, there is a reciprocal effect of gut microbiota on clock gene expression. Circadian oscillations of the gut microbiota result in oscillations of serum metabolites and are associated with transcriptional and epigenetic fluctuations in peripheral tissue [54]. Human and animal sleep-deprivation studies have reported that sleep disturbances have been associated with altered clock gene expression in humans, which vitally affects neurobiological responses to stress [55]. This chrono-disruption may sensitize individuals to stress and increase their vulnerability to stress-related disorders. Liang et al. discovered that the two primary components of the intestinal microbiota, Bacteroidetes and Firmicutes showed cyclical changes in abundance from day to night which are related to the biological clock and gender of the host [56]. Sleep disruption also resulted in altered faecal levels of bacterially modified metabolites including bile acids [57]. Interestingly, decreased melatonin levels have been proposed to mediate sleep deprivation-induced intestinal barrier dysfunction [58]. In addition, melatonin supplementation was able to attenuate sleep deprivation-induced dysbiosis as well as intestinal barrier dysfunction [58]. In addition to melatonin, group 3 innate lymphoid cells (ILC3s) are crucial mediators of circadian brain–gut signaling [59]. ILC3s express high levels of circadian clock genes and inversion of light–dark cycles leads to major circadian oscillations of ILC3s. This effect is dependent on the presence of *ARNTL* in the central nervous system (CNS) and the SCN of the hypothalamus and is furthermore associated with changes of gut microbiome composition, especially alterations in Proteobacteria and Bacteroidetes abundance [59].

Furthermore, the gut microbiota is directly involved in the production of a variety of neurotransmitters, cytokines and metabolites such as 5-HT, dopamine, gamma-aminobutyric acid (GABA), SCFA and melatonin. These metabolites act directly on the enteric nervous system and the vagus nerve and affect the activity of the CNS [60]. Importantly, some *Lactobacillus* and *Bifidobacterium* species can produce GABA. Abnormal expression of GABA mRNA is often observed in patients with depression and insomnia [61].

The vagus nerve is a major communication pathway between gut bacteria and the brain. This is of major importance, because the MGBA affects brain function through several pathways that produce a bidirectional flow of information [62]. The first is the immunoregulatory system in which the microbiota interact with immune cells affecting the levels of cytokines and prostaglandin E2 [63]. The second is the neuroendocrine pathway with more than 20 types of enteroendocrine cells in the intestine, which constitutes the largest endocrine organ. The gut microbiome affects the hypothalamic-pituitary-adrenal (HPA) axis and the CNS. The third is the vagus nerve pathway. The sensory neurons of the intestinal myenteric plexus are exposed to the gut microbiota through the regulation of the intestinal motility and gut hormone secretion. The intestinal nervous system also forms synaptic connections with the vagus nerve, which connects the intestine to the brain [64].

As summarized in the current introduction, sleep is a basic physiological requirement and indispensable for the regeneration of both body and mind. Many patients with psychiatric conditions such as affective disorders show disrupted sleep patterns as well as alterations of the gut microbiota.

(1)We aimed to review all cross-sectional studies investigating sleep and the gut microbiome in individuals with psychiatric disorders.(2)Furthermore, we aimed to systematically review all interventional studies, which measured sleep and the gut microbiome in individuals with psychiatric disorders.(3)Additionally, we aimed to narratively review the association between circadian rhythms, sleep and the gut microbiome in patients with psychiatric disorders and discuss the clinical implications of such associations, with particular emphasis on mechanisms of regulation of the MGBA and circadian rhythms in order to improve sleep (such as nutrition, probiotics and metabolites, psychotherapy and chronotherapy).

## 2. Methods

We performed a systematic review to analyze the interplay between disturbed circadian rhythms and the gut-microbiome only in psychiatric disorders (not in healthy controls). Search results were limited to publications in English from April 2015 to April 2020 due to the recent development of this field. We searched PubMed, Google Scholar, and Scopus for original articles analyzing the bidirectional cross-talk between sleep, circadian rhythms and the gut-microbiome in common psychiatric disorders using the following search terms: “sleep, microbiome and psychiatry”; “sleep, microbiome and depression”; “sleep, microbiome and bipolar disorder”, “sleep, microbiome and schizophrenia”, “sleep, microbiome and anorexia nervosa”, “sleep, microbiome and anxiety”; “sleep, microbiome and irritable bowel syndrome”, “clock gene expression and microbiome”, “clock gene expression and nutrition”. Reference lists of relevant articles were also reviewed to find additional literature Inclusion criteria were set a priori as follows: (1) cross-sectional studies analyzing sleep and gut-microbiome; (2) intervention studies analyzing sleep and gut-microbiome; (3) subjects had a diagnosis of depression, anxiety, bipolar disorder, schizophrenia, anorexia nervosa or irritable bowel syndrome (IBS) adhering to the Diagnostic and Statistical Manual of Mental Disorders (DSM-IV) diagnostic criteria to study. Exclusion criteria were: (1) systematic reviews or meta-analyses; (2) subjects with unaltered mental health; or (3) studies investigating mouse gut-microbiota. This systematic literature review was conducted in accordance with *PRISMA* recommendations [65]. The database search with the mentioned keywords above revealed *n* = 181 results. After careful consideration 29 were chosen for further investigation, but of these 29 articles, only three were identified that met inclusion criteria for this review (after exclusion of reviews investigating psychobiotics as treatment for anxiety, depression and related symptoms; sleep and nutrition interactions: implication for athletes; the role of the microbiome in insomnia; circadian disturbance and depression; study protocols or animal testing studies and analyses in healthy controls or other medical disorders). The workflow of the systematic review process is depicted in Figure 1.

Only clinical, randomized controlled trials (RCTs) and cross-sectional studies investigating sleep and the gut-microbiome published between April 2015 and 15 April 2020 including study participants with psychiatric disorders (BD, MDD, schizophrenia, anorexia nervosa, anxiety and IBS) were included in our systematic review. Cross-sectional and intervention studies analyzing circadian rhythms and the gut-microbiome in healthy controls were excluded. Similarly, animal studies investigating sleep and the gut-microbiome were excluded, but were later on included in the discussion. Case reports describing a connection between sleep and the gut-microbiome were also excluded based on the low number of participants (mostly single observations). Studies investigating subjects with unaltered mental health or no reported intervention were also excluded.

## 3. Results

Following a keyword search and elimination of excluded studies, 29 articles were found to discuss sleep and microbiome. Table 1 represents the searching results, which are depicted according to PICOS (population, interventions, comparisons, outcomes, study design) criteria and shows the final results of our systematic review, which includes only studies analyzing sleep and the gut-microbiome in psychiatric disorders in a cross-sectional, longitudinal or interventional design. Of these 29 articles only three studies were identified that met inclusion criteria (three cross-sectional studies analyzing sleep and gut-microbiome in individuals with depression and bipolar disorder and one interventional study analyzing sleep and gut-microbiome in individuals with IBS) [66,67,68].

Because of the focus on studies assessing persons with mental illness we excluded 12 systematic reviews or meta-analyses and 13 studies investigating healthy individuals or animals, which were discussed later on.

In Table 1, the findings of each study are depicted. Briefly, two studies investigated the gut microbiome in relation to illness severity including sleep of patients with bipolar disorder. One longitudinal case-control study from Evans et al. evaluated the gut microbiome composition in individuals with bipolar disorder in comparison to controls [66]. The study was conducted as a deep phenotyping longitudinal study with bi-monthly self-report measurements of physical and mental health. The authors compared the taxonomic composition of the gut-microbiome with self-reported disease measures. Cases were diagnosed with bipolar disorder according to the DSM-IV diagnostic criteria. Most of the patients were taking more than one psychiatric medication. The questionnaires, mailed every two months included the Patient Health Questionnaire-9 (PHQ-9), the Altman Self-Rating Mania Scale (ASRM), the Short Form Health Survey (SF-12), the Generalized Anxiety Disorder Assessment (GAD-7) and the Pittsburg Sleep Quality Index (PSQI). The stool microbiota sequences were binned into operational taxonomic units (OTUs) based on 97% sequence similarity using the average neighbor method. Analysis of molecular variance (AMOVA) between communities was used to determine if there were statistically significant differences between the microbiota from bipolar patients and controls. Quality 16S rRNA-encoding gene sequence data were generated from 233 patients with bipolar disorder and 179 controls were used in the analysis. Evans et al. found significant differences between the global microbiome communities and specific OTUs in individuals with bipolar disorder compared to controls. In the regression analyses of OTUs with self-report Sleep Quality Measures in patients with bipolar disorder there was a significant negative correlation (in the standardized ß coefficient from regression analyses) between self-report questionnaire scores and the outcome variable and microbial OTUs (*ß* = −0.329; *p* = 0.001; *d* = −1.357; *r* = −0.5) with *Faecalibacterium.* The gut presence of *Faecalibacterium* with a higher fractional representation could be associated with a healthier state. This study suggests that there may be a potential effect of *Faecalibacterium* on sleep quality. Methods to increase *Faecalibacterium* in the gut-microbiome may be important in regulating our biorhythm, a possible underlying mechanism being the production of butyrate which has sleep-enhancing properties [66].

The study of Aizawa et al. [67] examined the association between *Bifidobacterium* and *Lactobacterium* counts and affective symptoms in 39 patients with bipolar disorder (13 bipolar Type I and 26 bipolar Type II according to the DSM-IV) and 58 healthy controls using bacterial rRNA-targeted reverse transcription-quantitative polymerase chain reaction. Depressive symptoms were rated using the 17-item version of the Hamilton Depression Rating Scale (HAM-D) including the subscale for sleep, while the Young Mania Rating Scale (YMRS) was used to assess manic symptoms. Thirty-three of the patients were receiving pharmacological treatment. Bacterial counts in fecal samples were measured by using the Yacult Intestinal Flora-Scan based on 16S or 23S rRNA real-time quantitative polymerase chain reaction (RT-qPCR) to determine the composition of major gut bacterial groups. Comparisons of *Bifidobacterium* and *Lactobacillus* counts between the patients and controls revealed no significant differences between the two groups (*Bifidobacterium*: *df* = 1.92; *F* = 0.34, *P* = 0.56, *Partial η2* = 0.004; *Lactobacillus*: *df* = 1, 92; *F* = 0.14, *P* = 0.71) *Partial η2* = 0.002) In the patient group, there was no significant partial correlation (adjusted for age and sex) between bacterial counts and HAM-D total score (for *Bifidobacterium*: *ρ* = −0.06, *P* = 0.72; for *Lactobacillus*: *ρ* = −0.24, *P* = 0.16) or between bacterial counts and YMRS total score (for *Bifidobacterium*: *ρ* = 0.11, *p* = 0.53; for *Lactobacillus*: *ρ* = 0.25, *P* = 0.14). Subscales of the depressive symptoms (i.e., core, sleep, activity, psychic anxiety and somatic anxiety) were examined separately and a significantly negative correlation between *Lactobacillus* counts and sleep (*ρ* = −0.45, *P* = 0.01) was found. In this study no significant difference between patients with bipolar disorder and healthy controls were observed. However, a correlation between *Bifidobacterium* and *Lactobacillus* counts and depressive symptoms including disturbed sleep was found. In addition, there was a negative correlation between *Lactobacillus* counts and insomnia severity. Increasing *Lactobacillus* counts may be beneficial for sleep disturbances in bipolar disorder. One important limitation of this study is that most patients were treated with psychotropic medication and their disease severity was relatively mild (mean HAMD-D score of 10.3 ± 7.0 and YMRS score of 2.1 ± 3.5), which may have contributed to the lack of significant results [67].

Another study of Shunya Kurokawa et al. examined the effect of fecal microbiota transplantation (FMT) on psychiatric symptoms including sleep among patients with IBS functional diarrhea (FDr) and functional constipation (FC). Changes in HAM-D and sleep-related items, Hamilton Rating Scale for Anxiety (HAM-A) and Quick Inventory for Depressive Symptoms (QUIDS) were measured between baseline and four weeks after FMT. For donors, healthy relatives within the second degree of relationship (≥20 years of age) were screened using stool and serology screening for bacterial, parasitic, and viral pathogens. Seventeen patients were evaluated over the course of FMT to analyze the impact of FMT on psychiatric symptoms and to look for a relationship between microbiota composition and psychiatric symptoms. Fecal samples for microbiome analysis were longitudinally collected from patients at week 0, 1, 2 and 4 after FMT, and from donors on the day of FMT. Methods were fecal sample collection, bacterial DNA extraction, and 16S rRNA gene sequencing and analysis. The impact of FMT on sleep was evaluated using the total score from 3 HAM-D sleep-related scores. Mean depression and anxiety levels were assessed with HAM-D, HAM-A and QUIDS and found to be significantly decreased after FMT. Psychiatric symptom change before and four weeks after FMT including the sleep-related scores with standard deviation from HAM-D (sleep) 2.24 ± 2.22 to 0.64 ± 1.27 (*p* = 0.005; *FDR* = 0.007; *d* = 0.885; *r* = 0.405). Among all patients, the baseline Shannon index for diversity of intestinal microbiota and microbiota composition had a negative correlation with HAM-D total scores. The HAM-D score change was positively correlated with microbiome diversity change following FMT. The results of the study, assessing the effect of FMT on depressive, sleep and anxiety symptoms showed that FMT might be effective for these depressive symptoms. Shannon index indicated that the patients with HAM-D ≥8 showed significantly lower microbiota diversity compared to that of the healthy donors and patients with HAM-D <8. HAM-D score and microbiome diversity at baseline were thus, negatively correlated. The diversity changes after FMT significantly correlated with improvement of HAM-D scores [68].

## 4. Discussion

The aim of this review was to summarize the role of the gut microbiome in the context of circadian rhythms, nutrition and sleep disturbances in psychiatric disorders.

Out of 29 studies, three reports exploring the association between microbiome changes and sleep in psychiatric populations were identified. These studies showed that sleep disturbances may be associated with gut microbiome composition. In particular, sleep disturbance is a major symptom of many psychiatric illnesses and studies investigating the gut microbiome-sleep link in psychiatric populations are greatly needed. Three original reports exploring the association between microbiome changes and sleep in psychiatric populations were identified. Although these studies offer some insights into the potential relationship between the gut bacteria and the complex physiological process of sleep, much remains to be elucidated in this regard. The results of the current review are summarized and discussed. Psychiatric symptoms including depression, anxiety and sleep in patients with IBS may be improved after FMT [68]. The differences in improving insomnia severity may be due to improvement of patients’ GI symptoms and related depressive symptoms and can not be directly applied to patients with depression or anxiety disorders.

*Faecalibacterium* is a Gram-positive butyrate-producing member of the gut microbiota and may be beneficial in bipolar patients for reducing disease burden and improving sleep quality [66]. A higher fractional presentation of *Faecalibacterium* could be associated with a healthier state [69]. The intestinal microbiota and bacterial metabolites such as butyrate are likely to provide important links between the intestinal flora and sleep-generating mechanisms in the brain [70].

Methods to increase *Faecalibacterium* may include dietary approaches as described below. Targeting the microbiome may be an effective treatment paradigm for individuals with psychiatric disorders. There was a negative correlation between *Lactobacillus* counts and sleep [67]. *Lactobacillus casei* had beneficial effects on stress-induced sleep disturbance in healthy adults [71]. *Lactobacillus brevis* had beneficial effects on sleep rhythms in mice [72]. For the theoretical hypotheses and background as to why *Lactobacillus*, *Bifidobacterium* and *Faecalibacterium* are effective for sleep, as for their physiological increase, the impact of macronutrients on gut microbial composition is important. Plant-based protein has been associated with an increase in SCFA production, an improvement of gut barrier function and the beneficial modulation of the immune system by increasing regulatory T-cell expression and reducing inflammation. On the contrary, animal-based protein has been associated with an increase in trimethylamine N-oxide (TMAO) that contributes to cardiovascular disease progression and a reduction in SCFA production. Additionally, the quality of fat consumed is of importance for gut microbial composition which is in turn associated with adipocyte metabolism. Unsaturated fats appear to influence the growth of intestinal bacteria (e.g., *Lactobacillus, Bifidobacteria, Akkermansia muciniphila*) that are associated with reduced inflammation in white adipose tissue (WAT) and toll-like receptor (TLR) activation, along with lowering plasma cholesterol and LDL-cholesterol levels, whereas saturated fats are likely to support an increase in the abundance of *Bacteroides*, *Bilophila* and *Faecalibacterium prausnitzii* which may contribute to a reduction in insulin sensitivity and increased inflammation in WAT and TLR activation [43]. Predominantly plant-based diets, such as the Mediterranean diet, are characterized by large amounts of fruits, vegetables and grains which contribute substantially to the intake of prebiotic nutrients. There is a considerable body of evidence that adherence to a Mediterranean diet is beneficial for gut microbiota composition by improving the Bacteroides to Firmicutes ratio and leading to increased SCFA production. By contrast, the Western diet contains larger proportions of saturated fats, salt, and added sugars which negatively influences the gut microbiota composition by reducing *Bifidobacteria* abundance and butyrate-producing bacteria [73]. It has been proposed that cognitive impairment is linked to dysbiosis caused by the Western diet with possible mechanisms for this link being reduced SCFA availability in the gut, consequent impairments in gut barrier integrity, and elevated bacterial antigen and endotoxin influx in the circulation. This inflammatory condition is proposed to disrupt the blood brain barrier. Furthermore, SCFA mediate neurotransmitter secretion in the enterocytes and influence neuroendocrine pathways [74]. Valles-Colomer et.al. 2019 found a potential contribution of microbial GABA production (which is the main inhibitory neurotransmitter of the CNS well established that activation of GABA(A) receptors favors sleep) in depression and the gut microbes’ ability of the dopamine metabolite 3,4-dihydroxyphenylacetic acid (a dietary polyphenol that is predominantly contained in berries, fruits and vegetables) synthesis that is associated with higher perception of mental quality of life [75,76]. As summarized here, results in human study participants with psychiatric disorders are limited. However, studies involving healthy individuals and animal models are promising, and relevant findings will be discussed in the following paragraphs.

### 4.1. Clock Genes, Diet and Sleep Disturbances Affect Physiological Oscillations of the Gut Microbiome

Although studies involving human subjects are limited, the relationship between the molecular clock and microbiome has been analyzed in a variety of animal studies. Alongside other influences, feeding rhythms and diet composition are major factors influencing diurnal oscillations of the gut microbiota [77,78]. Chronic sleep fragmentation of mice has been reported to increase the Firmicutes to Bacteroidetes ratio as well as the relative abundance of members of Lachnospiraceae and Ruminococcaceae, while decreasing the relative abundance of Lactobacillaceae families [35]. SCFAs also impact clock genes, given that oral administration of SCFAs cause phase changes in the peripheral clock in mouse peripheral tissue [79]. In addition, the SCFA butyrate has been demonstrated to increase non-rapid-eye movement sleep (NREMS) in rodents [80].

### 4.2. Gut Microbiome Diversity Is Associated with Sleep Physiology in Healthy Individuals

Sleep and microbiome changes have been extensively investigated in healthy individuals. In novel bioinformatic analyses, such as redundancy analysis, co-occurrence analysis and artificial neural networks, it was demonstrated that the composition, diversity and metabolic function of the gut microbiota was significantly changed between those with insomnia and the healthy population. Random forest together with cross-validation identified two signature bacteria, which could be used to distinguish insomnia patients from the healthy population (*Bactecteroides; Clostridiales*). The gut flora interaction networks were significantly altered for patients with insomnia compared with the control group [81]. Smith et al. used actigraphy to quantify sleep measures coupled with gut microbiome sampling to determine how the gut microbiome correlates with various measures of sleep physiology. They found that total microbiome diversity was positively correlated with increased sleep efficiency and total sleep time along with positive correlations between total microbiome diversity and interleukin-6. Their analysis of the microbiome composition revealed that phyla richness of Bacteroidetes and Firmicutes were positively correlated with sleep efficiency and several taxa like Lachnospiraceae, *Corynebacterium* and *Blautia* were negatively correlated with sleep measures [82]. These quantified sleep measures differ clearly from the self rating measures described in the reviewed papers in patients with psychiatric diseases and clarify the need of objective measures in patients with psychiatric diseases. Jason R. Anderson suggested a possible relationship between sleep quality, gut microbiome composition and cognitive flexibility in healthy older adults. Regarding the quality and duration of sleep, an observational study showed a positive association with fruit and vegetable intake—which are the major dietary suppliers of prebiotic fibers and polyphenols. Young men with higher consumption of fruit and vegetables had better sleep quality and shorter time to fall asleep [83].

### 4.3. Clinical Implications

#### 4.3.1. The Interplay of Dietary and Nutritive Composition, Microbiota, Mental Health and Sleep Quality

Psychobiotics have the potential to be a novel and well-tolerated treatment strategy in MDD, but only a few placebo-controlled probiotic trials have been performed in patients with psychological symptoms. Akkasheh and colleagues conducted a randomized, double-blind, placebo-controlled clinical trial with 40 MDD patients to investigate the effect of probiotics capsules containing *Lactobacillus acidophilus*, *Lactobacillus casei* and *Bifidobacterium bifidum*. They discovered beneficial effects using measures of the Beck Depression Inventory (BDI), insulin levels and inflammation markers after eight weeks of probiotic intake [84]. Grünwald et al. analyzed 42 males and females suffering from stress and exhaustion, which improved after an RCT with a probiotic multivitamin preparation (*L.acidophilus* and *B. bifidum* and *B. longum*) [85]. In the study of Yu et.al., milk was fermented with the strain *Lactobacillus brevis* DL1-11 for which high GABA-producing capacity had been proposed. The effects on anxiety and sleep quality were evaluated in mice that were administered different doses of the GABA-containing fermented milk. Additionally, significant increases in *Ruminococcus*, *Adlercreutzia* and *Allobaculum* abundance and elevated levels of the SCFA butyric acid were observed. Thus, GABA-fermented milk may improve sleep by changing gut microbiota and SCFA levels [86].

Of note, most of the probiotics RCT studies only included healthy individuals, and were summarized by an excellent comprehensive review by Wallace and Millev [87]. In Figure 2 we roughly outline the important relationships between the molecular clock, the MGBA and nutritional psychiatry as future clinical implication. Thus, further research is necessary before starting the era of “psychobiotics” in psychiatric treatment. Currently, there are only few studies published that explicitly investigated microbiota-shaping ability of diet and its interplay with sleep quality. Thampson et.al focused on the potential impact of dietary prebiotics and bioactive milk fractions on sleep quality in a rat model. In rats that were fed a diet rich in prebiotics, lactoferrin and milk fat globule membranes, increased *Lactobacillus rhamnosus* abundance, improved NREM sleep and REM sleep quality after acute stressor exposure were noted, as well as a more stable gut microbial alpha diversity and diurnal rhythms [88].

#### 4.3.2. Dietary Habits and Sleep

Besides the possible impact of the gut-microbiota on sleep quality, eating habits may also affect sleep quality in a bidirectional manner.

(1)Sleep habits affect how much and what we eat

Sleep potentially modulates metabolic function, including energy metabolism and appetite. Pot et.al comprehensively reviewed the potential contribution of sleep duration on dietary choices. A reduced sleep duration was associated with poorer food choices and increased dietary and energy intake. However, energy expenditure is not equally elevated, leading to a positive energy balance and, in the long-term, weight gain. The changes in energy intake associated with sleep deprivation were accompanied by an unfavorable nutrient distribution, a limited variety of food choices and consequently potential state of gut dysbiosis. Thus, sleep has been proposed as a possible modifiable risk factor for chronic non-communicable diseases [89]. A negative association of fruit and vegetable intake and sleep habits was also reported in the study of Jansen et.al [83].

(2)Dietary patterns and habits affect how we sleep

Additionally, dietary patterns can also impact sleep [89,90]. Higher fat intake is associated with disordered sleep whereas the Mediterranean diet is associated with fewer sleeping problems. Additionally, skipping breakfast, irregular eating habits, low intake of vegetables and fish, and high intake of simple carbohydrates (such as sweets and noodles) have been associated with poor sleep quality in Japanese females [89].

#### 4.3.3. Psychotropic Medication Affecting Circadian Rhythms

Even though the reviewed topic belongs to a new area of psychiatric research, there has long existed a diverse range of psychotropic medications concerned with the remediation of disturbed circadian rhythms. Mood stabilizers such as lithium and valproate, both commonly used in the treatment of bipolar disorder, modulate circadian rhythms via inhibition of glycogen-synthase-kinase-3-beta (GSK3Beta), which phosphorylates the cycle length determining *PER* clock gene products. Lithium also inhibits the clock component REV-ERBα that inhibits *ARNTL* itself [91,92,93]. However, there are very few studies investigating the effect of lithium and valproate on the MGBA, Furthermore, new drug targets in the “sleep–gut axis” could potentially be of vital importance in addressing the widespread sleeping difficulties seen in psychiatric disorders.

#### 4.3.4. Psychological and Psychotherapeutic Interventions

In clinical practice, several psychological interventions to improve sleep quality are well evaluated and described: (1) sleep hygiene, (2) cognitive-behavioural therapy (CBT), and (3) relaxation, mindfulness and hypnotherapy [94].

In the psychological management of sleeping disorders, psychoeducational approaches and day structure interventions are also highly common and effective and aim to change circadian rhythms and improve psychological conditions including sleep quality [95]. An example is social rhythm therapy (SRTs) which focuses on stabilizing the daily activities (e.g., meal schedule, sleep/wake patterns, activity times), which could have a positive impact on the circadian system. Subsequently, the patient learns to structure the day cognitively and to plan possible changes in routines. Patients learn to adapt to the social rhythms of the environment and to correct irregularities in life habits. In particular, in bipolar disorder, interpersonal and social rhythm therapy (IPSRT) is well validated [96,97]. However, recent research highlights the positive effects of IPSRT on cognitive functioning in patients with major depression [98]. Additionally, manipulating the sleep–wake cycle seems to be a promising intervention in the clinical management of major depression and bipolar disorder [99,100].

## 5. Conclusions

Our systematic review confirmed that there are important crosslinks between disturbed circadian rhythms and the MGBA, which can affect well-being and symptomatology throughout the course of psychiatric disorders. This is an innovative approach in attempting to elucidate and understand disease mechanisms and has diverse clinical implications. However, currently, studies investigating the microbiome–sleep link in patients with psychiatric disorders are sparse. We could identify only four studies analyzing the bidirectional crosstalk between sleep and the gut microbiome in human study participants with psychiatric disorders. Future studies on a multi-omics level are necessary to analyze the interaction between circadian rhythms with regard to the gut microbiome, metabolomics, genomics and transcriptomics. Further integrative studies using machine-learning approaches in a holistic way are important to elucidate the sleep and the MGBA as the “via regia” to a harmonized biorhythm and wellbeing.

## Figures and Tables

**Figure 1 nutrients-12-02198-f001:**
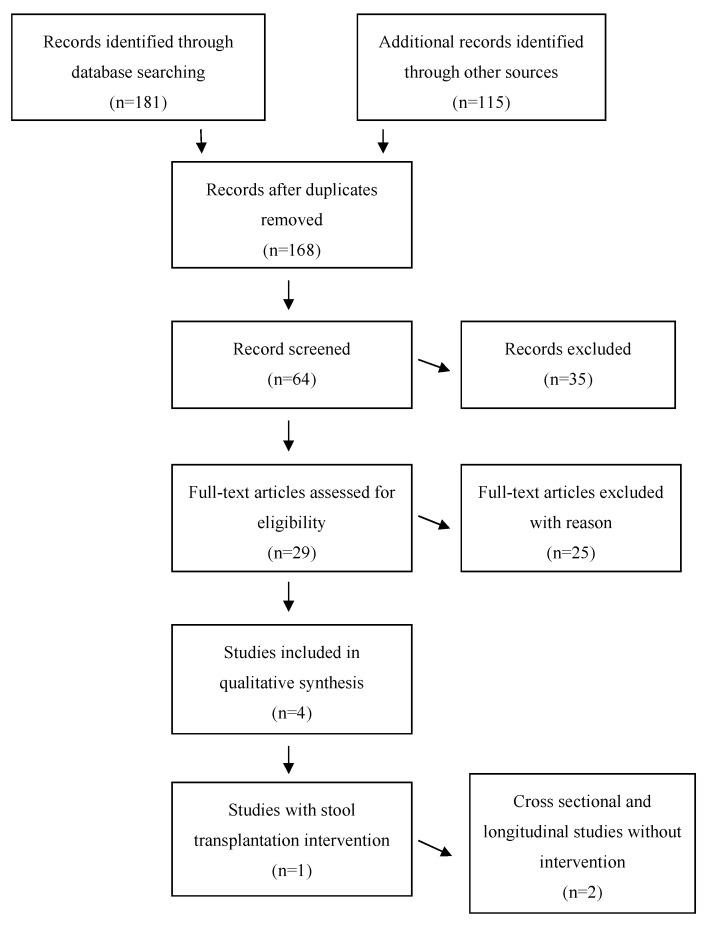
Flow diagram of article screening, selection and elimination of the systematic database search.

**Figure 2 nutrients-12-02198-f002:**
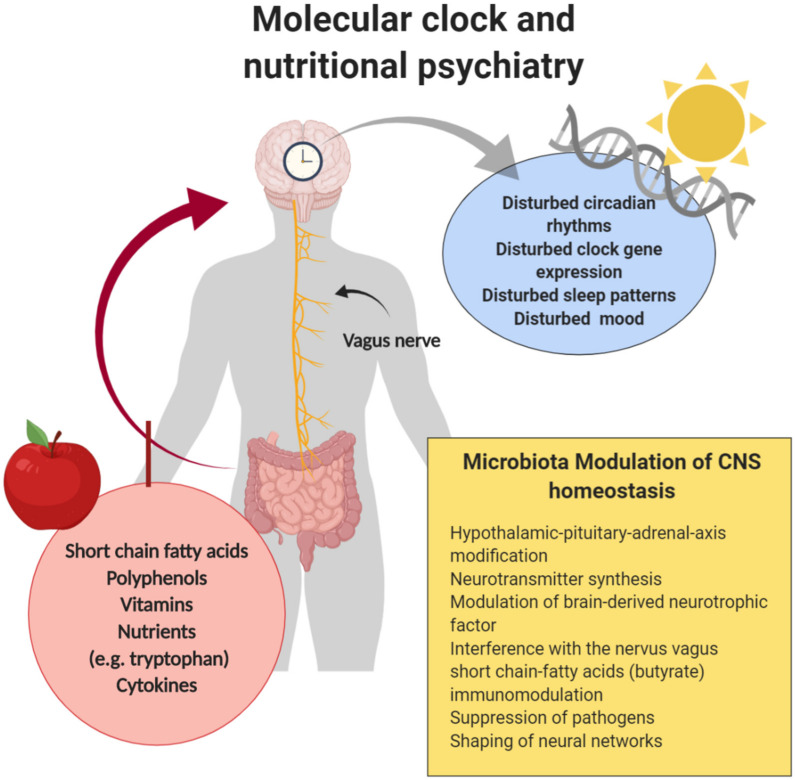
Molecular clock, microbiota–gut–brain axis (MGBA) and nutritional psychiatry. The figure was designed with Biorender.com.

**Table 1 nutrients-12-02198-t001:** Human studies published between 2015 and 2020 that investigated circadian rhythm and microbiome in psychiatric disorder and irritable bowel syndrome. DSM = Diagnostic and Statistical Manual of Mental Disorders; OUT = taxonomical unit-level analysis; Pittsburg Sleep Quality Index (PSQI); Short Form Health Survey (SF-12); Generalized Anxiety Disorder Scale (GAD); The Altman Self-Rating Mania Scale (ASRM); Food Frequency Questionnaire; Young Mania Rating Scale; HAM-D = Hamilton Rating Scale for Depression; HAM-D Subscale Sleep; IBS-QOL = Irritable Bowel Syndrome Quality of Life questionnaire; HADS-A = Hospital Anxiety and Depression Scale-Anxiety; HADS-D = Hospital Anxiety and Depression Scale [66,67,68].

**Case Control Studies Investigating Sleep and Microbiome in Psychiatric Disorders**				
**Author and Year of Publication**	**Population**	**Measurements/Interventions**	**Comparison**	**Outcome**
Evans et al., 2016	Individuals with bipolar disorder	Pittsburg Sleep Quality Index (PSQI), Operational Taxonomical Unit level analysis (OTU), molecular variance (AMOVA), Short Form Health Survey (SF12), Generalized Anxiety Disorder scale (GAD), Altman Self-Rating Mania Scale (ASRM)	Bipolar Disorder (n = 115) healthy controls (n = 64); cross-sectional study	Negative relationship between *Faecalibacterium* and sleep quality (PSQI) at the subscale level (*ß* = −0.329; *p* = 0.001; *d* = −1.357; *r* = −0.5)
Aizawa et al., 2019	Individuals with Bipolar Disorder	Bacterial counts (RT-qPCR) Hamilton Depression Rating Scale (HAM-D), Young Mania Rating Scale	Bipolar Disorder (13 bipolar Type I, 26 bipolar Type II), 58 healthy controls; cross-sectional study	Negative correlation between *Lactobacillus* counts and sleep. (*p* = −0.45, *P* = 0.01; *d* = 0.885; *r* = 0.405)
**Intervention Studies Investigating Sleep and Microbiome in Psychiatric Disorders**				
**Author and Year of Publication**	**Population**	**Interventions**	**Comparison**	**Outcome**
Kurokawa et al., 2018	Individuals with Irritable Bowel Syndrome (IBS)	Fecal Microbiota Transplantation, Hamilton Rating Scale (HAM-D), Subscale of sleep related items, Hamilton Rating Scale for Anxiety (HAM-A), Quick Inventory for Depressive	17 Patients with Irritable Bowel Syndrome (IBS), functional diarrhea FDr, functional constipation FC, DSM IV of Comorbid depression and anxiety, Intervention Study, single center, open-label, non-randomized observational study	Significant improvement in HAM-D total and sleep- subscale score. Baseline Shannon index show lower diversity in patients with HAM-D > 8 compared with patients HAMD < 8 (*p* = 0.005, *FDR* = 0.007)

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
