# Peer review of "Sleep and Microbiome in Psychiatric Diseases"

_nutrients, 2020, doi:10.3390/nu12082198_

Round 1

Reviewer 1 Report

The subject and search in the present review article was interesting. However, a few references were not cited. It was critical that the study of Kurokawa et al. (2018) was not put in References. Other some references were not in References but in text.

Description of Figure 2 was not in the text.  

Reviewer 2 Report

Review

Wagner-Skacel et al report review about the role of the gut microbiome in the context of circadian rhythms, nutrition and sleep in psychiatric disorders. The adaptation of living organisms in their environments depends significantly on their ability to develop biological rhythms, which control the body functions starting from the cellular level up to behavior. According to the prevailing views of scientists today, the dark/light cycle plays a leading role in the organization of biosystems. The authors summarized in this research the current introduction, sleep is a basic physiological requirement and indispensable for the regeneration of both body and mind. Many patients with psychiatric conditions such as affective disorders show disrupted sleep patterns as well as alterations of the gut microbiota. The gut microbiota significantly influences the functioning of both the digestive and immune systems of the host and even affects the peripheral and central nervous systems via the gut-brain axis. Previously the host molecular clock has been shown to influence the composition of the gut microbiota.

The results obtained by the authors are very interesting and useful; however, the presentation and discussion of the results could be improved. This application should be of interest to a quite broad audience, however, some issues listed below make should be corrected. A lot of typos on the text. 

Abstract: Should be rewritten. The methods must be removed from the abstract. It is necessary to summarize the works considered in the review and make a general conclusion.

Introduction

Throughout the text, the taxa should be written in italics.

Line 55-56 Please, add reference to the following description.

Line 81 The molecular 24-hour clock. The main proteins of the mammalian molecular clock can be divided into two groups: those expressed upon exposure to light (BMAL1 and CLOCK) and those expressed in the dark phase (PERs and CRYs). While the mRNA coding for BMAL, PER and CRY proteins experience oscillations in the SCN depending on the circadian phase, the concentration of CLOCK protein remains stable at all times. Other proteins regulating the circadian clock influence the expression of crucial genes, which are known as the clock-controlled genes. In this section, it is necessary to say about PERs and CRYs too.

Line 143-144 The number of fecal bacteria as well as the abundance of Bacteroidetes oscillate during the light-dark cycle in mice. What does fecal bacteria mean?

Line 164-165 Repeat lines 129-130.

Line 175 the gene.

Line 177-178 Please, add reference.

Results 

Line 282-283 Please, add reference.

Line 284-285 Please, add reference.

Line 318-319 abbreviations must be defined (RT-qPCR)

Line 388 Table 1: add references to these articles.

Discussion

Line 411 gut microbiomecomposition change for diversity of gut microbiome.

Line 516 B. longum

Reviewer 3 Report

This manuscript is targeting a very stimulating issue. It is however much too long as it sometimes takes the reader into circles and dead ends so that the interest loosens up. The introduction, for example, could be tighten up:

- The historical background does not bring together useful knowledge to the rest of the manuscript.

- The section on Neurophysiology of sleep describes techniques of quantified EEG analysis that are not used elsewhere in the manuscript.

- The section on molecular 24-hour clock does not serve any purposes neither as no data on circadian rhythms is discussed later on.

- The Gut microbiome and sleep section is more focused on chronobiology and biochemistry than sleep itself.

Given the paucity of papers surviving the exclusion criteria, it might have been useful to include other reviews and papers on animals or healthy humans in order to further the understanding the possible interaction between “Sleep and Microbiome in Psychiatric Diseases”. The scientific substance gets a little thin with 4 papers on 3 different diagnoses, one of which is GI-related.

The section results

Please identify clearly from the start which are the four papers to be analyzed and provide the full reference.

- The first paper (Evans et al, line 284), gets fully cited at the very end of the paragraph (line 308).

I think the conclusion on its analysis I a little bit farfetched: “therapeutically increasing Faecalibacterium, for example, by dietary approaches in bipolar patients may be beneficial in reducing disease burden”. It should be clearly mentioned if it is a citation from the original authors themselves. If it is from the authors of this manuscript, I would suggest some caution. It is not because a substance is missing in a diseased organism that it will normalize by providing supplementary doses of this substance. The link between Sleep and Microbiome in Psychiatric Diseases is most probably more complex that  that.

- The second paper is Chung et al (no year cited, line 309) and it is not in the reference list. It is cited in Table 1 as “Aizawa et al., 2019”. I searched it and found the following: Chung YE, Chen HC, Chou HL, et al. Exploration of microbiota targets for major depressive disorder and mood related traits. J Psychiatr Res. 2019;111:74-82. This looks like reference #80 (but the year is 2018) and it does not report sleep data…

The discussion on this paper is significantly shorter than for the three others.

- The third paper is that of Aizawa. It is not included in the reference list. It is cited in Table 1 as “Chung et al., 2018”. I searched it and found the following: Aizawa E, Tsuji H, Asahara T, Takahashi T, Teraishi T, Yoshida S, Koga N, Hattori K, Ota M, Kunugi H. (2019). Bifidobacterium and lactobacillus counts in the gut microbiota of patients with bipolar disorder and healthy controls. Frontiers in Psychiatry, 9, 730.

This paper reports on bipolar disorder and therefore should have been presented after that of Evans et al (2017), also on bipolar disorder.

On line 336 the authors of the manuscript report on sleep duration while no duration data is reported in the original paper since they used qualifiers from the HAM-D scale on insomnia severity.

- The fouth paper is that of Kurokawa et al. It is not included in the reference list. It is cited in Table 1 as “Kurokawa et al., 2018”. I searched it and found the following: Kurokawa et al. The effect of fecal microbiota transplantation on psychiatric symptoms among patients with irritable bowel syndrome, functional diarrhea and functional constipation: An open-label observational study. Journal of Affective Disorders Volume 235, 1 August 2018, Pages 506-512.

The authors of this review manuscript should discuss whether sleep is improved because GI symptoms improved or because depression improved.  

The cited reference at the end of the paragraph (line 364) is a Pubmed ID number.

Discussion

- Line 309: four or three papers? Does Chung et al really show sleep data?

- Section 4.1: There is nothing in the data that relates to clock genes. I would drastically cut down on this section.

- Section 4.2 discuss on sleep in healthy persons. First, it calls for including papers on healthy persons in this systematic review. Second, you discuss quantified sleep measures, sleep efficiency and total sleep time. These are objective measures while the reviewed papers are reporting HAM-D and PSIQ scores, which are subjective complaints: not the same at all. This should at least be discussed.

In my opinion the rest of the discussion is much too long and runs away from the central topic. The dietary aspects are at least two steps away from the literature analyzed, so is the discussion on circadian rhythms and on psychotherapy.

Minor points

The writing style is sometimes anecdotal:

- Mentioning “Nobel Prize Winners”, authors’ forenames, “handful of studies” (l. 178)

Conclusion: It is clear that great efforts have been put into writing this review. However, I find that parts of it should belong to another paper.
